# The Positive Cycle of Breastfeeding—Mental Health Outcomes of Breastfeeding Mothers Following Birth Trauma

**DOI:** 10.3390/healthcare13060672

**Published:** 2025-03-19

**Authors:** Abigail Wheeler, Fay Sweeting, Andrew Mayers, Amy Brown, Shanti Farrington

**Affiliations:** 1Psychology Department, Faculty of Science and Technology, Bournemouth University, Poole BH12 5BB, UK; fsweeting@bournemouth.ac.uk (F.S.); andymayers92@gmail.com (A.M.); sfarrington@bournemouth.ac.uk (S.F.); 2Public Health, School of Health and Social Care, Swansea University, Swansea SA2 8PP, UK; a.e.brown@swansea.ac.uk

**Keywords:** breastfeeding, postnatal care, mental health, birth trauma, qualitative

## Abstract

Background/Objectives: It is established that both birth and infant feeding experiences can impact maternal mental health, but little is known about how the two might interact. Potentially, a positive breastfeeding experience might help to mitigate feelings associated with birth trauma, but conversely, a difficult feeding experience might further compound birth trauma. The aim of this study was therefore to explore how mothers’ experiences of breastfeeding following birth trauma might impact their mental health. Methods: To explore this, mothers were invited to complete an online survey about their experiences of birth and breastfeeding and how they felt these affected their wellbeing. There were 501 responses, with 159 (32%) describing their birth experience as traumatic and therefore included in this analysis. A thematic analysis approach was used to explore themes around breastfeeding experiences and the impact on wellbeing. Results: Mothers who described positive breastfeeding experiences felt that breastfeeding helped them to bond with their baby post birth trauma and that this protected their mental health. Conversely, mothers who had a difficult experience described how pain, exhaustion, and low milk supply further negatively impacted their wellbeing. Conclusions: The findings show the importance of enhanced support for breastfeeding mothers who have experienced birth trauma, as feeding experiences can potentially help heal or compound challenging memories, thoughts, and emotions around birth.

## 1. Introduction

Perinatal mental health refers to mothers’ wellbeing from conception up to one year after birth [1]. The Royal College of Midwives (RCM) and the Maternal Mental Health Alliance highlight that perinatal mental health conditions affect around 20% of women during pregnancy and in the first year after giving birth, although this number may be underestimated due to women sometimes concealing their challenging emotions [2]. Understanding triggers for perinatal mental health issues is important to support mothers, babies, families, and the economy. The costs of treating perinatal mental health are in excess of £74,000 for one mother and child [3].

One trigger for distress during the perinatal period is birth experience. Birth trauma can be defined as distress experienced after going through or witnessing a traumatic birth [4]. It is thought that 15.7% of mothers will experience some trauma-related symptoms following birth, with 4–6% developing post-traumatic stress disorder (PTSD) [5]. Women experiencing birth trauma can have long-lasting symptoms such as flashbacks, severe anxiety and distress, and physical symptoms. However, difficult emotions such as feeling like a failure or a ‘bad’ mother or struggling to bond with their baby are common [6]. Trauma during birth can be affected by medical complications for the mother or infant such as haemorrhage, pain, and tearing, forceps or emergency c-section delivery, or a neonatal care stay.

Mothers experiencing birth trauma have to work through their experience whilst also caring for their new baby. The transition to motherhood and caring for a dependent infant is recognised as being challenging, especially when many new parents are isolated and living away from family support. The feelings of birth trauma can make caring for a baby feel even more challenging, but it is likely that experiences of infant care might further exacerbate birth trauma [7]. One potential important factor is infant feeding experience.

Breastfeeding and mental health can be closely linked. When mothers want to breastfeed and breastfeeding goes well, this can be supportive of mental health, leading to feelings of increased confidence, positive maternal identity, and satisfaction. However, when breastfeeding difficulties arise or mothers need to stop breastfeeding before they are ready, this can lead to feelings of failure, alongside grief, loss, and anger if she then must stop breastfeeding [8]. Unfortunately, in the UK, many women do not meet their infant feeding goals due to an under investment in infant feeding support combined with little societal understanding of how to support breastfeeding and new parents more broadly [9]. Researchers investigated the bidirectional relationship between mental health and breastfeeding with 109 UK mothers across 24 interviews. There was a mixed impact of breastfeeding on mental health with the following themes: failure to breastfeed, the impact of ceasing breastfeeding, and the positive impact of positive breastfeeding experiences. The researchers concluded that the bidirectional relationship consisted of negative breastfeeding experiences impacting mental health, along with poor mental health impacting negatively on breastfeeding experiences [10].

Breastfeeding, birth trauma, and difficulties can be closely linked. A difficult birth experience is associated with lower rates of breastfeeding initiation and continuation due to physiological and psychological reasons. Birth interventions can negatively impact the hormones needed for milk production, whilst residual pain and exhaustion can make breastfeeding more difficult. They can also reduce feelings of maternal confidence and self-belief in her ability to feed her baby [11].

Postnatal PTSD is associated with lower breastfeeding rates, as shown in a systematic review of 21 studies [12]. Studies in the review showed that mothers with postnatal PTSD did not breastfeed for as long as they intended to. One Israeli study showed that a large number of mothers who experienced postnatal PTSD did not breastfeed their infants for 6 to 8 weeks post birth [12]. This demonstrates the impact of trauma on breastfeeding ability, as mothers consistently demonstrated lower breastfeeding initiation and duration rates. In one study with Chinese mothers, 759 eligible participants completed the Posttraumatic Stress Disorder Checklist (PCL-C) at 42 days post birth and answered a series of questions about their breastfeeding experience [13]. Results reported that 92 participants had developed PTSD 42 days after birth, and 4% of those exclusively breastfed, while 14.5% of them partially breastfed (in combination with formula, cow’s milk, etc.). The researchers found that mothers who exclusively breastfed had a substantially lower risk of experiencing postnatal PTSD. Supporting these findings, researchers demonstrated that women who had postnatal PTSD were six times less likely to initiate breastfeeding compared to mothers who did not have postnatal PTSD. They also found that maternal depression was associated with the non-initiation of breastfeeding, as well as ceasing breastfeeding before 1 year postnatally [14]. This may indicate the negative impact of trauma and wellbeing on breastfeeding experiences. The current study aims to extend these findings.

General PTSD symptoms have been compared with birth-related ones with regard to mother–infant bonding. Researchers reported that while general PTSD symptoms negatively impacted the bond, birth-related PTSD symptoms did not [15]. This requires further study but may highlight the differences in how trauma impacts bonding. In another study, researchers explored how breastfeeding difficulties impact mother–infant bonding. Finding that, on average, mothers who reported difficulties with breastfeeding reported lower bonding than mothers who did not report breastfeeding difficulties [16]. To our knowledge, there is little research into the impact of birth trauma and breastfeeding on bonding; the current study aims to look at this. There is also little research on the positive accounts of mothers who have experienced a traumatic birth, which this study also aims to explore.

We therefore know that both birth and breastfeeding experiences can affect maternal mental health, but there has been little consideration of how the two may interact with each other. We know that a difficult birth experience can make breastfeeding more difficult, likely exacerbating perinatal mental health difficulties. However, it is possible that when breastfeeding goes well, the experience may have a positive impact on maternal confidence, bonding, and wellbeing. This study aimed to explore how subsequent infant feeding experiences affected maternal wellbeing in a sample of women who had experienced birth trauma.

## 2. Materials and Methods

### 2.1. Study Design

The data used in this study were collected as part of a larger study exploring the perceived pressures and mental health of breastfeeding mothers. An online survey consisting of open-ended questions was completed by mothers between January and February 2021, and the data were re-analysed in 2023 for this study, and the data pertaining to birth trauma, breastfeeding, and mental health outcomes were selected.

### 2.2. Setting and Sample

Using social media resources (Facebook and Instagram), and the personal social media account of one of the authors (A.M.), 926 participants were recruited through convenience sampling. Inclusion criteria included being over the age of 18 and having given birth prior to the COVID-19 pandemic, all participants gave informed consent to take part in this study. A total of 667 participants met the inclusion criteria, and upon removing any incomplete and duplicate responses, 501 remained. The current study included only analysing the data from mothers who had shared that their birth experience was traumatic; thus, only 159 responses have been used for the analysis here. The mean age (at birth) of these participants was 30.7 years old, ranging from 17 to 49 at the time of giving birth. The year of birth ranges from 1985 to 2020, and this wide date range allows for a variety of perspectives and an understanding of breastfeeding over time; however, it is acknowledged that this could impact the relevance of the data.

### 2.3. Data Collection

Data were collected using social media between January and February 2021. Breastfeeding organisations were also used to advertise the study for data collection, and they did so by sharing the survey link with their networks. Participants gave informed consent and were aware that they could choose to withdraw from this study at any point, and this study was designed online using Qualtrics ^TM^. Informed consent was imbedded into the Qualtrics ^TM^ programme and participants had to consent before moving forward to the survey questions. If participants did not give consent, they were debriefed at that stage and provided with support resources.

Organisations and experts in the field were approached to review the questionnaire (which was initially proposed by A.W. and A.M.) and to ensure content validity before the final study (see Table 1). The focus of the questions asked was the experiences of infant-feeding mothers, covering how they fed their infant, the challenges with this, and the impacts on their mental health—the survey also asked whether they considered their birth to have been traumatic. No formal scales were used to evaluate the trauma responses; instead, these were self-reported by mothers in response to the question, “Would you consider that birth to have been traumatic?”. Trauma was a theme which emerged from the original dataset and therefore was analysed further in this paper. The age of the mother and the year of birth were also collected. Participants received a debrief and were also signposted to supporting organisations.

### 2.4. Ethical Considerations

The questions used in the survey were designed with inputs from experts in the field, to ensure that it was worded in a manner to support engagement and avoid any distress in the participants who may have experienced any trauma during their birth. Participants were informed that they had a choice to withdraw from this study at any time and had been signposted to information of national and local support, if needed, to manage any distress they may have experienced.

The data were always stored anonymously for analysis, and any personally identifiable data were separated at the point of data collection and complied with both the GDPR and university regulations (see Section A.1).

### 2.5. Data Analysis

The current study used Reflexive Thematic Analysis (RTA) [17,18], utilising NVivo 20 to analyse the data to generate the broad themes and subthemes (the method used was identical to a previously published study) [19]. A.W initially generated the codes and remained reflexive at each stage. Codes were initially grouped to form themes (A.W.), then reviewed by the other authors (A.W., A.M., S.F., and F.S.), they were then clearly defined. The initial results were drafted by A.W and then edited by the remaining authors (A.M., S.F., F.S., and A.B.). Reflexive Thematic Analysis values the subjectivity of the researcher into the analytical process, and the trustworthiness of the researcher is essential in the analysis process to demonstrate credibility, transferability, dependability, and confirmability [20]. The researchers continue to engage with experts, strive to increase their knowledge, and remain self-aware to mitigate any potential bias.

### 2.6. Trustworthiness

Trustworthiness is essential to qualitative analysis and often demonstrated by valuing credibility, transferability, dependability, and confirmability. Diverse participant recruitment sources, such as social media, demonstrate credibility in this study. The researcher is constantly reflecting on their decisions, potential biases, and thought processes throughout the use of RTA, which in turn maintains the credibility of the findings.

Transferability is gained through the responses from the varied participants who engaged in this study. The context is visible to the readers; therefore, the responses can be generalised to other settings and have practical implications in this context.

Dependability in this research has been demonstrated through code re-coding, which was carried out by the lead researcher after the initial coding was completed using RTA, revisiting and revising the codes and ensuring that the interpretations align with the context of the main research question, and was validated by the other researchers in this study.

Reflexivity was the key aspect throughout the analysis process, ensuring minimal confirmability bias. The researcher continuously reflected on their own positionality to minimise the impact this could have on data interpretation. The use of direct participant quotes ensures that participant voices are accurately portrayed and represented.

### 2.7. Author Positionality

RTA as a method acknowledges the bias that researchers may have, especially in the context of qualitative data analysis. A.W., the lead researcher, has been aware of her position as a woman who has not personally experienced childbirth and breastfeeding. Her interpretation of this research is subjective based on her limited personal experiences, and she recognises that her knowledge and understanding comes from secondary sources such as friends, family, support groups, and reviewing research. These sources have enabled her to increase her knowledge on breastfeeding and childbirth and will have influenced this project. The other authors have their own experiences and contributed to the validation of the analysis during the research.

Considering this when using RTA, the data were initially coded based on emerging themes and re-coded once potential biases were considered. The final codes are the product of a collaborative effort between authors, who all have their own individual experiences, which contributed to the credibility and validation of the analysis.

## 3. Results

A total of 159 participant responses were included in this study; Table 2 shows the location and the participant’s ages, with a mean age of 30.5.

Emerging from the data, the themes were divided into the positive and negative impacts on mental health. Under the umbrella of negative impacts, the subthemes included pain, low milk supply, exhaustion, and postnatal depression. While under the positive impacts, the subthemes included bonding and promotion and mental health, as seen in Figure 1.

### 3.1. Overarching Mental Health Impact

Of the 159 mothers who reported that their birth experience was traumatic, there were 125 positive references and 280 negative references in relation to emotional wellbeing, breastfeeding, and mental health. Positive references highlighted where successful breastfeeding had contributed to emotional wellbeing and improved bonding, whereas negative references referred to challenges faced when breastfeeding, as well as how these experiences compounded the already traumatic birth and impacted mothers’ wellbeing negatively.

It is important to recognise the impact that birth trauma can have on other aspects of caregiving, especially engaging with breastfeeding. This study explores how trauma can have both positive and negative influence on breastfeeding, which in turn may directly affect the overall wellbeing of birth mothers, including postnatal depression (PND).

### 3.2. Negative Impacts on Mental Health

Negative impacts were sometimes attributed to physical difficulties when breastfeeding, such as pain, exhaustion, and low milk supply, while others shared that they struggled emotionally, for example, experiencing feelings of anxiety when it came to ensuring their baby was fed, and a sense of failure or guilt due to their breastfeeding challenges. Following this, some mothers also shared that they received a diagnosis of postnatal depression.

### 3.3. Postnatal Depression

There were 25 mentions of postnatal depression, and some mothers attributed their postnatal depression (PND) to a lack of support with their breastfeeding journey. Some mothers received treatment for this, such as talk therapy, but many had to battle for any support relating to their wellbeing.

“I had PN depression, later diagnosed with generalised anxiety disorder when my youngest was 6 months old. Had intensive CBT [cognitive behavioural therapy] course face to face, met once a week” (participant 159, Ref 10, gave birth in 2018 at age 29 by emergency c-section to their third child, still breastfeeding at 2 years).

“When it was so hard at the beginning and I didn’t get listened to or the support I needed, I felt so desperate and it definitely contributed to my postnatal depression” (participant 195, Ref 11, gave birth in 2017 at age 35 by vaginal assisted delivery to their first child, breastfed for 19 months).

“Horrible. Lack of support, undiagnosed tongue tie, pressure from bf cafe and midwife to NOT use formula at all costs. I was diagnosed with PND and put my BF [breastfeeding] experience as one of the leading factors” (participant 292, Ref 17, gave birth in 2015 at age 27 by vaginal assisted delivery to their first child, breastfed for 22 months).

### 3.4. Low Milk Supply

Some mothers struggled with low milk supply, which made breastfeeding a challenge both physically and emotionally. A lack of knowledge and information was a key factor, as some mothers reported not receiving information on why their milk supply was low, for example, due to an emergency caesarean or premature birth. Following an already traumatic birth experience, a low milk supply left some mothers feeling helpless (not having any control over this or awareness) and had a further negative impact on their wellbeing.

“The insufficient supply of milk and how that made me feel as a mother”. (Participant 170, Ref 5, gave birth in 2017, aged 29, by emergency c-section to their first child, and breastfed for 18 months).

“A slow start—hospital claimed my milk wasn’t sufficient so I was forced to top up with formula after a traumatic birth” (participant 204, Ref 7, gave birth in 2018 at age 31 by vaginal delivery to their first child, breastfed for 2 years).

“Awful. Baby was 5 weeks premature, and milk hadn’t come in, this wasn’t explained. We stayed in hospital for 5 days after birth (I hemorrhaged after birth and had to be resuscitated), baby kept screaming, I kept trying to feed and nothing” (participant 316, Ref 18, gave birth in 2015 at age 27 by vaginal delivery to their first child, breastfed for 28 months).

### 3.5. Exhaustion 

Feelings of tiredness/exhaustion were also a source of difficulty during some mothers breastfeeding journeys. For some, breastfeeding was not easy, leaving the mothers with feelings of inadequacy and exhaustion. Others expressed that exhaustion often left them feeling overwhelmed at the responsibility of being the only person able to feed their infant. Being the sole caregiver to an exclusively breastfed infant meant that there was no one else who could support them to feed, which many felt was a huge weight on their shoulders.

“Initial feelings of exhaustion of inadequacy that breastfeeding was not straightforward” (participant 17, Ref 2, gave birth in 2018 at age 35 by vaginal delivery to their first child, still breastfeeding at 25 months).

“The exhaustion from all night feeding and at times feeling overwhelmed with baby constantly on top of me” (participant 339, Ref 20, gave birth in 2017 at age 27 by vaginal assisted delivery to their first child, still breastfeeding at 3 years and 10 months).

“Exhausting at times and felt like all the responsibility on me as no one else could feed my baby so in the early days felt a huge responsibility” (participant 129, Ref 7, gave birth in 2020 at age 31 by vaginal assisted delivery to their first child, still breastfeeding at 5 months).

### 3.6. Pain

Out of all the subthemes under the negative impacts, pain was the most reported difficulty, with 43 references expressing this. Pain often led mothers to question whether they could continue with their breastfeeding journey, whether they could push through the pain. In some cases, this impacted their wellbeing significantly, leaving them under stress and feeling down.

“I was broken and ready to give up completely and in a lot of pain” (participant 16, Ref 6, gave birth in 2013 at age 30 by vaginal delivery to their first child, breastfed for 1 year).

“Initially the latch-on was incorrect and very painful. There were moments I wasn’t sure I could continue, and the idea of feeds was very stressful”. “There were some evenings I just cried whilst trying to get through the pain of feeding along during the night” (participant 199, Ref 20, gave birth in 2017 at age 20 by emergency c-section to their first child, breastfed for 2 months).

### 3.7. Positive Impacts on Mental Health

From 159 mothers who experienced a traumatic birth, there were 125 references to positive experiences relating to their breastfeeding journey. Some felt that positive breastfeeding experiences supported and promoted their mental health—often this was attributed to healing following a traumatic birth. Many mothers also shared that the bond with their infant was positively impacted by a successful breastfeeding experience, again with an attribution to healing after trauma.

### 3.8. Promotion of Mental Health

There were 58 references to breastfeeding promoting mental health in mothers who had experienced birth trauma. Some mothers attributed breastfeeding as a support, helping them heal after the birth trauma, and some mothers felt very strongly about how breastfeeding had treated their mental health. Some even attributed breastfeeding to saving their life; these are powerful statements in support of breastfeeding being an effective tool for treating poor mental health following birth trauma.

“It had a positive impact after having such a traumatic birth, I knew I wanted to breastfeed for as long as I could and it really helped me heal” (participant 70, Ref 8, 28 years old at the time of birth in 2018, emergency c-section, second child, 12.5 months).

“I think it was exceptionally important. I don’t think I would’ve recovered from the experience of the birth as well if I hadn’t breastfed. I genuinely felt at one point that this was the only thing I could do for her” (participant 99, Ref 13, gave birth in 2016 at age 29 by emergency c-section to their first child, breastfed for 14 months).

“Brilliant, once we were able to breastfeed, I strongly believe that this was almost the cure to my poor mental health. Being successful and providing for my daughter meant so much and made me look after myself too” (participant 357, Ref 41, gave birth in 2018 at age 32 by vaginal assisted delivery to their first child, breastfed for 3 years and 6 months).

“Knowing that she was only fed my milk even through a tube created a connection when I couldn’t even hold her. And it helped us to bond when she was finally discharged. It was a very important part of emotional healing and did a lot for my mental health.” (participant 202, Ref 17, gave birth in 2019 at age 32 by emergency c-section to their first child, breastfed for 19 months).

### 3.9. Bonding

This was the largest subtheme overall, with 67 mothers sharing this impact. Mothers shared that bonding with their infant during breastfeeding gave them a strength and purpose. Some mothers expressed how a positive breastfeeding journey supported their bond and helped them treat or prevent their postnatal mental health difficulties.

“I suffered severe mental health issues for a few years before having my baby and breastfeeding was a huge help in grounding me and bonding with my baby” (participant 91, Ref 12, gave birth in 2019 at age 17 by vaginal delivery to their first child, still breastfeeding at 14 months).

“I think after the hideous start we had, it helped me to connect with him and feel that I could do something right I think it helped me recover from the trauma and I think pumping milk gave me a purpose while he was in ITU [intensive therapy unit]” (participant 478, Ref 69, gave birth in 2018 at age 32 by emergency c-section to their first child, breastfed for 5 months).

“It helped me build a bond with my daughter after a traumatic birth and whilst I dealt with ppd [postpartum depression]” (participant 121, Ref 17, gave birth in 2016 at age 25 by vaginal delivery to their first child, breastfed for 15 months).

“Extremely beneficial to my bonding with baby and my mental health. I think I would have suffered PND if not for the success and bonding I was able to experience through breast feeding” (participant 320, Ref 51, gave birth in 2019 at age 33 by emergency c-section to their first child, breastfed for 2 years).

## 4. Discussion

This study aimed to explore the breastfeeding experiences of mothers who had also reported a traumatic birth, and the impact of this on their perinatal mental wellbeing. Overall, findings suggest a link between birth trauma and perinatal mental health difficulties. The finding with the most impact was the strong interaction between positive breastfeeding experiences and how this improved perinatal mental health. The reports from mothers who attributed being able to breastfeed successfully with a positive effect on their mental health, in some cases labelling it ‘lifesaving’, are very powerful. A cycle of how positive breastfeeding experiences lead to a positive impact on wellbeing following birth trauma is emerging, and to our knowledge, this has not been previously reported. Further research is required to confirm these findings, but it is hypothesised that supporting mothers to breastfeed successfully could potentially be used as an effective immediate treatment for poor perinatal mental health, especially following birth trauma. This is especially important considering that 20% of new and expectant mothers experience perinatal mental illness [2], 15.7% of mothers experience some trauma-related symptoms following birth, and 4–6% develop diagnosed PTSD [21]. In previous research, it has been concluded that resource provision to support women with birth trauma experiences within the NHS is not sufficient, and they attributed this to a lack of funding, recording systems, and publicity—they also found that midwives were not receiving specialist training to support mothers [22]. Our data suggest that a solution may be to empower perinatal health professionals to use positive breastfeeding experiences to improve maternal wellbeing. For example, more specialist training for midwives could be initiated, as well as more resources to support awareness around some of the difficulties with breastfeeding and to support mothers who have experienced a traumatic birth by implementing positive breastfeeding strategies, thereby effectively using existing resources.

Our data indicated that pain was the most reported physical difficulty amongst mothers who had also experienced birth trauma. Mothers expressed the upset and stress that this caused, with some unsure if they could continue as they wished with breastfeeding. This supports existing research, which found that pain levels were positively correlated with depression levels and lower self-efficacy [23]. We have also reported how other physical difficulties such as exhaustion and tongue tie impacted perinatal mental health, supporting a recent study which showed that higher fatigue and poor sleep quality increased mothers’ risk of postnatal depression [24].

Studies have found that postnatal PTSD is associated with lower breastfeeding rates, and maternal depression was linked to the non-initiation of breastfeeding [12,20]. Both studies reported that maternal mental illness could be treated by supporting mothers to breastfeed successfully. Building on this premise, data from our study indicate that positive breastfeeding experiences could be used to treat birth trauma related PTSD. This supports findings from another study, which reported that mothers who exclusively breastfed had a substantially lower risk of experiencing postnatal PTSD [13]. Evidence also suggests that the COVID-19 pandemic impacted breastfeeding support and may have led to early cessation and lower rates of breastfeeding [25]. While the current study excluded mothers who had given birth during the pandemic, it is recognised that the pandemic may still have impacted the levels and quality of breastfeeding support.

Bonding was a strong theme that emerged from the positive experience of breastfeeding in this study when there was birth trauma, contradicting the results where trauma could lead to poorer bonding but supporting the findings that successful breastfeeding rates can support this bonding experience [16]. Mothers in this study alluded to the fact that successful breastfeeding saved their mother–infant bond following a traumatic birth, and this is a powerful sentiment which further adds to notions of the positive cycle of breastfeeding.

### Limitations

This study focused on experiences prior to the COVID-19 pandemic; however, data were collected in 2021, over a year beyond the beginning of the pandemic. Therefore, recall bias is a potential limitation, as the quality of the memory and recall of the participants could be questioned, with some experiences being many years prior to data collection. However, in a recent study, it is suggested that mothers’ memories of birth experiences are more vivid and reliable, particularly in relation to birth trauma. The researchers concluded that birth memories tend to be consistent, unfragmented, and central to women’s lives [26].

The data were filtered based on the participants perception of their birth experience being traumatic. The participants were not given parameters regarding what trauma is defined as; therefore, this was based on their own idea of what defines trauma. This is a potential limitation, as each participant’s knowledge of trauma would differ. However, trauma is inherently subjective and therefore, even with defined parameters, this could still have been a limiting factor.

## 5. Conclusions

Overall, this study fulfilled its aims of exploring the breastfeeding experiences of mothers who reported a traumatic birth and how this impacted their perinatal mental wellbeing. The findings demonstrate that birth trauma and subsequent breastfeeding experiences have a profound impact on mental health. Negative impacts on mental health were linked to physical difficulties, such as pain and exhaustion, and lack of support received. On the other hand, positive impacts were demonstrated where breastfeeding was promoted and was shown to improve mental health and bonding outcomes. The emergence of the positive breastfeeding cycle suggests that effective breastfeeding support could be a protective factor for mothers’ mental health following birth trauma. Further research should explore this idea with the purpose of improving breastfeeding and perinatal support services for mothers.

## Figures and Tables

**Figure 1 healthcare-13-00672-f001:**
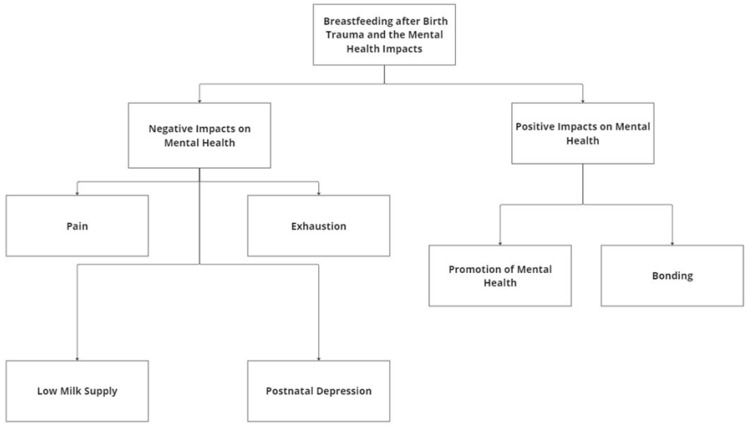
A coding tree showing themes and subthemes.

**Table 1 healthcare-13-00672-t001:** A table showing the summary of key questions.

Summary of Key Questions
Can you tell us about your breastfeeding experience?
How did those around you feel about you breastfeeding?
Did you encounter any difficulties with breastfeeding?
To what extent did breastfeeding promote your mental health?
What aspects of breastfeeding had a negative impact on your mental health?
Did you receive any support with breastfeeding? If so, from whom?
What further support would you have liked?
What support did you get for your mental wellbeing during this period?
What further support would you have liked for your mental wellbeing?
Is there anything that you feel would have improved your breastfeeding experience in any way?

**Table 2 healthcare-13-00672-t002:** A table showing participant demographics.

Location of Birth	Number of Mothers
Hospital or Birthing Clinic	154
Home	1
Home but Transferred to Hospital	4
**Type of Birth**	
Vaginal (Assisted)	46
Vaginal (Spontaneous)	44
Vaginal (Induced)	7
C-Section (Planned)	51
C-Section (Emergency)	11
**Mothers Age at Birth**	
17–20	8
21–25	17
26–30	50
31–35	60
36–40	21
41+	3

## Data Availability

The data pertaining to the analysis of this study are available from Abigail Wheeler upon reasonable request.

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
