# Peer review of "The Positive Cycle of Breastfeeding—Mental Health Outcomes of Breastfeeding Mothers Following Birth Trauma"

_healthcare, 2025, doi:10.3390/healthcare13060672_

Round 1

Reviewer 1 Report

Comments and Suggestions for Authors

The main question addressed by the research is How did mothers experience their breastfeeding journey as it relates to their wellbeing - what were positive and negative factors in their experiences. The themes and experiences add to the literature as they relate to a specific subset of women who had traumatic birth experiences. 

The data was collected in Jan-Feb 2021, but the survey asked for respondents who had given birth before the pandemic - so that was over a year prior to the time the data was collected.
Is that potentially a problem since people are describing past situations? Could the pandemic have clouded some of the memory of their postpartum journey. 

Where there parameters around what was considered a "traumatic birth" for data collection purposes?

As I understand it, reflexive thematic analysis is iterative, can the authors speak more about the iterative process of navigating their biases and then how they came back to data and so on. What was learned through exploring their biases.

Is this the type of data that necessarily needs reflexive thematic analysis? The researchers did not interview, they are coding written responses so it seems like the role of researcher bias might actually be minimal?

 Are the conclusions consistent with the evidence and arguments presented
and do they address the main question posed? Please also explain why this
is/is not the case.

I'm stuck on "positive" and "negative" - how did the team sort these out. Those terms feel very full of value judgement, and I want to make sure the researchers did not impose external values into what is positive or negative. 

Other minor suggestions :

line 178- should minimal

line 184 - AW, the lead researcher, ....

line 186 - change "my"

line 224 - is it talking therapy or just "talk therapy"

Interesting study and well written

Author Response

Dear Reviewer,

Thank you very much for your comments and valuable feedback on our manuscript. We appreciate the time and effort you have taken to review our work, and your suggestions have been incredibly helpful in strengthening the paper.

We have carefully considered each of your comments and have made revisions accordingly. Below, we provide a detailed response to each point, outlining the changes made and the rationale behind them. Within the manuscript, changes are also highlighted in red text.

Kind Regards,

The Authors

Comment 1: The main question addressed by the research is How did mothers experience their breastfeeding journey as it relates to their wellbeing - what were positive and negative factors in their experiences. The themes and experiences add to the literature as they relate to a specific subset of women who had traumatic birth experiences. – Thank you for this comment

Comment 2: The data was collected in Jan-Feb 2021, but the survey asked for respondents who had given birth before the pandemic - so that was over a year prior to the time the data was collected.
Is that potentially a problem since people are describing past situations? Could the pandemic have clouded some of the memory of their postpartum journey. – Thank you for your comment, we recognise the potential for recall bias in this study when asking participants about past situations. While this is a potential bias, the lasting emotional impacts of giving birth are often profound. AltuntuÄŸ et al (2024) found that women’s birth memories tend to be consistent, unfragmented, and emotionally significant a year or more after giving birth. The study found that 45.9% of women had a high or very high perception of traumatic childbirth, and this perception explained 17.3% of birth memories and recall. These findings suggest that trauma-related birth memories are particularly vivid.

We have added a limitations section within the discussion to highlight this potential bias (see line 410) we have also added research regarding the impacts of Covid-19 on breastfeeding support into the discussion (see line 397).

Comment 3: Where there parameters around what was considered a "traumatic birth" for data collection purposes? – there weren’t, and this is due to the fact the data was originally collected as part of different study, and later re-analysed. The original question which this data was filtered from was: “Would you consider that birth to have been traumatic?” therefore trauma was something that the participant determined independently based on their own perception. It is especially important to capture what women thought of as traumatic, rather than have a defined parameter for the study.

We have included a few sentences regarding this in the “Data Collection” section (see line 146) we have also acknowledged this in the limitations section (see line 418)

Comment 4: As I understand it, reflexive thematic analysis is iterative, can the authors speak more about the iterative process of navigating their biases and then how they came back to data and so on. What was learned through exploring their biases. – thank you for your comment, the data analysis process was done in stages. Data was first coded based on what stood out initially, and this was then considered in relation to personal research biases – eg the lead researcher is a woman who has never given birth or breastfed, therefore her perceptions of birth trauma and breastfeeding experiences may differ from the other researchers, or indeed people who have experienced birth trauma and breastfeeding. To address this, the data was re-coded with these potential biases in mind and then reviewed by the other researchers to ensure a more balanced interpretation. The final codes are the product of this processes of initial coding and personal reflection on personal influences.

We briefly describe the lead authors positionality in section 2.7, however we have added further detail on how this related specifically to RTA (see line 204)

Comment 5: Is this the type of data that necessarily needs reflexive thematic analysis? The researchers did not interview, they are coding written responses so it seems like the role of researcher bias might actually be minimal? – thank you for your comment, while we understand that RTA is best suited to interview data, the sensitive and subjective nature of birth trauma and breastfeeding experiences, we felt it was the most appropriate analysis technique. We actively reflected on potential biases throughout the coding process, and this enabled us to ensure the final codes actively represented the experiences of the wide range of participants we had. 

Comment 6: Are the conclusions consistent with the evidence and arguments presented
and do they address the main question posed? Please also explain why this
is/is not the case. - We agree that it is important to ensure the conclusions are consistent with the evidence and arguments presented. Our conclusion highlights that the study successfully met its aim of exploring the breastfeeding experiences of mothers who reported a traumatic birth and examining the impact of these experiences on their perinatal mental well-being. The findings demonstrated a profound impact on mental health, with negative effects linked to physical difficulties with breastfeeding and positive effects observed when breastfeeding was successfully supported.

We have made some revisions to the conclusion section to make this clearer.

Comment 7: I'm stuck on "positive" and "negative" - how did the team sort these out. Those terms feel very full of value judgement, and I want to make sure the researchers did not impose external values into what is positive or negative. – thank you for your comment, we understand how there could be a risk of value judgement when considering whether responses were “positive” or “negative”. This is where the iterative and reflexive nature of RTA supports coding and reduces the risks of value judgement. Initial codes emerged indicatively, based on participants own language and descriptions. Researcher biases were then reflected on to support reducing these impacts on the data. When considering positive and negative elements, this was done using the participants own language as a guide. For example, negative codes included words and phrases such as “horrible”, “lack of”, “inadequate” which are inherently negative. While positive codes included words and phrases such as “helped”, “beneficial”, “important”. This approach ensured that themes were a true reflection of the participant’s experiences, rather than external value judgements.

Other minor suggestions :

Thank you for the below suggestions, these have been corrected in the manuscript.

line 178- should minimal

line 184 - AW, the lead researcher, ....

line 186 - change "my"

line 224 - is it talking therapy or just "talk therapy"

Interesting study and well written - thank you, we really appreciate your comments

AltuntuÄŸ K, Kiyak S, Ege E. Relationship between birth memories and recall and perception of traumatic birth in women in the postpartum one-year period and affecting factors. Current Psychology. 2024;43(1):876-84.

Reviewer 2 Report

Comments and Suggestions for Authors

Check in the text all the abbreviations and control if there were reported in extenso the first time that appears in the text (e.g. line 222 “post natal depression”; at line 226 “I had PN Depression”) and if were reported correctly (e.g. Line 235 “I was diagnosed with PND..”).

  • Lines 75-77. Please add the reference of the systematic review cited.
  • Line 117. Could you specify what do you mean with : “e data was re-analysed in 2023 for this study”? Maybe you mean that within the information collected in 2021, data on birth trauma and breastfeeding were selected for the analysis in 2023?
  • Line 126. At line 126 the renge of age reported is 17-49; how many woman with 17 years old were included? In the inclusion criteria (line 121) is specified that the participants shlud have 18 years old. Please clarify this issue.
  • Lines 119 and 131. Please reported the social media used (e.g. instagram, facebook or others) and what do you mean with “the network of one of the authors”. The social media refers to an institutional account or a personal account? It is not clear.
  • Line 131. Breastfeeding organisations involved used our istitutional account to disseminate the survey, or provide the invitataion with email or links?
  • Line 133. The informed consent was provided before the start of the survey? It is integrated in the Qualtrics link? Or it was presented by other ways?
  • Line 146. Please delate the second “experience” term (“…during their experience with childbirth.”) in this sentence, to avoid repetition.
  • Line 186. Please correct the sentence, because is write in third person but there are reference in first person (e.g.” …based on my limited personal experiences..”).

Comments on the Quality of English Language

Please check all the typo in the text (example: line 150 “annoymusly” with “anonymously”).

Author Response

Dear Reviewer,

Thank you very much for your comments and valuable feedback on our manuscript. We appreciate the time and effort you have taken to review our work, and your suggestions have been incredibly helpful in strengthening the paper.

We have carefully considered each of your comments and have made revisions accordingly. Below, we provide a detailed response to each point, outlining the changes made and the rationale behind them. Within the manuscript, changes are also highlighted in red text.

Kind Regards,

The Authors

Comment 1: Check in the text all the abbreviations and control if there were reported in extenso the first time that appears in the text (e.g. line 222 “post natal depression”; at line 226 “I had PN Depression”) and if were reported correctly (e.g. Line 235 “I was diagnosed with PND..”). – thank you for this comment, we have gone through the paper and added any relevant abbreviations. The majority of these are in the direct quotations from participants.

Comment 2: Lines 75-77. Please add the reference of the systematic review cited. – thank you for your comment, we have moved the systematic reference forward to make this clearer within the text (see line 76)

Comment 3: Line 117. Could you specify what do you mean with : “e data was re-analysed in 2023 for this study”? Maybe you mean that within the information collected in 2021, data on birth trauma and breastfeeding were selected for the analysis in 2023? – thank you for this comment, while true that data were selected for the analysis in 2023, the data was also re-analysed in full. We have added a line to make this clearer in the manuscript (see line 117)

Comment 4: Line 126. At line 126 the renge of age reported is 17-49; how many woman with 17 years old were included? In the inclusion criteria (line 121) is specified that the participants shlud have 18 years old. Please clarify this issue. – we can confirm that all participants were over the age of 18 at the time of participating in the study, however the age range refers to the age they were when they gave birth, which could have been under the age of 18.  We have added a sentence to make this clearer (see line 128)

Comment 5: Lines 119 and 131. Please reported the social media used (e.g. instagram, facebook or others) and what do you mean with “the network of one of the authors”. The social media refers to an institutional account or a personal account? It is not clear. – thank you for your comment, I have added the social media account type and platforms used (see line 120)

Comment 6: Line 131. Breastfeeding organisations involved used our istitutional account to disseminate the survey, or provide the invitataion with email or links? – the breastfeeding organisations involved shared our survey link with their networks. We have provided clarity on line 134

Comment 7: Line 133. The informed consent was provided before the start of the survey? It is integrated in the Qualtrics link? Or it was presented by other ways? – thank you for your comment, informed consent was imbedded in the Qualtrics link, before being able to proceed with the survey, the participants read the study information, which included information about the right to withdraw, all aspects of the survey and why the study was being run. Participants had to provide consent to move forward to the survey questions, if they did not provide consent, they were passed to a debrief page thanking them for considering taking part and given links to support resources. We have made this clearer in the manuscript (see line 137)

Comment 8: Line 146. Please delate the second “experience” term (“…during their experience with childbirth.”) in this sentence, to avoid repetition. – thank you for highlighting this, we have removed the repeated word.

Comment 9: Line 186. Please correct the sentence, because is write in third person but there are reference in first person (e.g.” …based on my limited personal experiences..”). – thank you for highlighting this, we have amended this (see line 198)

Comment 10: Please check all the typo in the text (example: line 150 “annoymusly” with “anonymously”). – thank you for highlighting this, we have amended this (see line 161)